# Design and Validity of a Choice-Modeling Questionnaire to Analyze the Feasibility of Implementing Physical Activity on Prescription at Primary Health-Care Settings

**DOI:** 10.3390/ijerph17186627

**Published:** 2020-09-11

**Authors:** Sergio Calonge-Pascual, Francisco Fuentes-Jiménez, José A. Casajús Mallén, Marcela González-Gross

**Affiliations:** 1ImFINE Research Group, Faculty of Physical Activity and Sport Sciences-INEF, Universidad Politécnica de Madrid, C/Martín Fierro 7, 28040 Madrid, Spain; s.calonge@upm.es (S.C.-P.); francisco.fuentes@upm.es (F.F.-J.); 2GENUD Research Group, Faculty of Health Sciences, University of Zaragoza, Pedro Miral s/n, 50008 Zaragoza, Spain; joseant@unizar.es; 3Biomedical Research Center of Physiopathology of Obesity and Nutrition, CIBERobn, (CB12/03/30038), Carlos III Health Institute, Avenida de Monforte de Lemos 5, 28029 Madrid, Spain

**Keywords:** exercise, preventive medicine, health promotion, public health, physicians, nurses, health behavior, healthcare, attitude of health personnel, disease management

## Abstract

Worldwide health policies are trying to implement physical activity on prescription (PAP) at healthcare settings. However, there is not a proper methodology to analyze PHC organizational staff factors. This study aims to validate two questionnaires to assess the self-perception of nurses and general practitioners to implement PAP at primary healthcare (PHC) settings. The designed choice-modeling Google-form questionnaire was sent to 11 expert nurses and 11 expert sports medicine physicians. Experts evaluated each question on a 1–5 points Likert-type scale according to their expertise. Aiken’s V coefficient values ≥0.75 were used to validate separately each question using the Visual Basic-6.0 software. A total of 10 sports medicine physicians and 10 nurses with 28.4 ± 5.1 y and 16.3 ± 11.8 y of PAP experience, respectively, validated the questionnaire. One expert in each group was not considered for offering 3 ± SD answers in ≥2 questions respect to the mean of the rest of experts. Final Aiken’s V coefficient values were 0.89 (0.77–1.00) for the nurses’ questionnaire and 0.84 (0.77–0.95) for the physicians’ one. The questionnaires designed to assess the PAP self-perception of PHC nurses and physicians were validated. This methodology could be used to analyze PHC organizational staff factors in order to achieve an efficient PAP implementation in other PHC contexts.

## 1. Introduction

The global age-standardized prevalence of physical inactivity is around 25% in the world [1]. Physical inactivity and sedentary behavior represent currently a leading risk factor for non-communicable chronic diseases (NCDs) [2]. Since, in 2015, Saltin et al. established that exercise prescriptions could treat at least 26 different NCDs [3], physical activity (PA) has increased in importance and represents a cornerstone in the prevention, at least, of 35 chronic conditions [2]. Recently, it has been published that even more than 40 NCDs could be prevented and treated by exercise prescriptions as a precision medicine [4]. Health benefits of exercise prescriptions are documented and well-known [5]. Therefore, several approaches are trying to promote PA on prescription (PAP) in healthcare (HC) settings, as for example, the proposal of the WHO for the European Region 2016–2025 [6], the worldwide strategy of the American College of Sport Medicine that is the Exercise is Medicine^®^ initiative [7,8,9], and others [10,11,12].

Based on the strength of existing evidence, regular PA should be the first line of preventive and rehabilitation medicine; however, insufficient progress is being developed about the implementation of a model to increase PAP in HC settings [10,13,14]. Interest in PAP implementation has increased during last years, and adherence to non-pharmacological treatments had been traditionally focused by patient-related factors [15]. In contrast, some studies have shown the lack of public resources in the HC system [16,17,18] and for the HC team [16,19], the lack of exercise prescription training knowledge [20,21,22,23], lack of time in their Primary Healthcare (PHC) consultations [24], and the lack of awareness with exercises prescription [24,25], etc.

PHC providers should be considered as intra and interpersonal dimensions inside of the organizational HC system dimension under a social ecological perspective on PAP programs [26,27]. The relationship with the local health and sports community resources [28], managed by health promotion policy, should be strategically studied in order to developed a PAP strategy by a multidisciplinary or interdisciplinary approach [12]. In this sense, to identify all the main barriers and facilitators that influence PAP in HC settings should be a priority for an efficient implementation [10,17]. Some attempts have been made in the last years. Surveys have been designed and carried out in physicians from Canada [7,29], Germany [30,31], and USA [32,33], in health promoters from Australia [34,35] and physical therapists from Canada [36] and Nigeria [37]. However, there is not a consensus or a validated questionnaire to measure the self-perceived PAP barriers and facilitators in health promoters. Therefore, as part of a broader study, the aim of this study was to validate two questionnaires by a panel of experts who were familiar with the PAP construct, to assess the self-perception barriers and facilitators of nurses and general practitioners (GPs), as a first step to facilitate the implementation of PAP at PHC settings, under the umbrella of the Exercise is Medicine initiative.

## 2. Materials and Methods

Two questionnaires were validated according to the guidelines published by Tsang et al., in 2017 [38]. Following these guidelines, domains of interest and constructs were previously identified to assess what the questionnaire should measure and a check that no validated questionnaire is available in the scientific literature was performed. The baseline for the questionnaire design were focus groups discussions with 5 GPs and 5 nurses, which took place as part of a PhD thesis [12]. Briefly, GPs and nurses, separately, guided by an expert psychologist, commented during four hours about their opinions, barriers, and facilitators to implement PAP in PHC. Their answers were categorized and organized in a structured 30-question questionnaire, according to the five sections and category and subcategory constructs shown in Table 1.

Two choice-modeling, Google-form questionnaires were developed by our group [12] as an on-line self-administered format, because we thought that health professionals are able to complete the questionnaire on their own. A close-ended item format by multiple-choice, Likert-type scales and true/false answers were established, with a simple, short, and familiar language item design for the target respondents. As there is no rule for the length of questionnaires, it was designed to measure the full construct of interest, trying not be so long that respondents experienced fatigue or loss of motivation in completing it. A preliminary review and revision pool of items and pilot testing was conducted by all members of the Improvement of health by fitness, nutrition and exercise Research Group (ImFINE).

Finally, a content validity of the questionnaires was done by a panel of experts as previously had been done by other authors in other fields [39,40]. Eleven nurses and 11 GPs were selected. Inclusion criteria for being considered as an expert were: age ≥ 35 y; >3 y of PAP career experience and/or ≥20 y of academic background related to sports medicine, general practitioner medicine, nursing, public health promotion, and recent or previous relationship with a national public health or PHC system. Descriptive data of the experts are shown in Table 2 and Table 3.

The link to the questionnaires (nurses: https://forms.gle/CmJDQAjR5Pt1zLp36; sports medicine physicians https://forms.gle/coQttEgtBPYgH7Qj7) was sent via email, previous consent to participate. Experts had to indicate below each question their degree of agreement (1–5 points in the Likert-type scale, where 5 points indicated the highest agreement and 1 the lowest agreement). When an expert’s opinion was ≥3 ± standard deviation (SD) different from the mean of the other 9 experts in two or more questions, these values were not considered valid because of the discordance with the rest of experts. The coefficient of content validation for the final 10 experts was calculated using Aiken’s V coefficient (95% CI) [40]. This coefficient and the lower and upper confidence intervals were calculated using the free software ICaiken.exe (Visual Basic 6.0, Lima, Perú) [40]. A minimum Aiken’s V coefficient score of ≥0.75 was needed for each question to be validated [39].

The study was performed according to the principles established with the Declaration of Helsinki 1964 and further amendments and other national regulations for research projects involving human participants: Protection of Personal Data, Law 15/1999 of 13 December on the Protection of Personal Data provided in the current legislation (Royal Decree 1720/2007 of 21 December). The protocol study was approved by the Ethical Committee of the “Hospital Universitario Fundación Alcorcón” and the Central Commission for research of the Region of Madrid (RP1811600040).

## 3. Results

A total 10 sports medicine physicians and 10 nurses with a mean of 28.4 ± 5.1 y and 16.3 ± 11.8 y in PAP experience, respectively, were finally considered for the validation of the questionnaires.

The 30 items of each questionnaire were agreed by each expert from 1 to 5 points, being 5 the maximum agreed value. The sports medicine physician expert 3 (Table 4) and the expert 7 of the nurses’ questionnaire (Table 5) were not considered for offering 3 ± SD answers in ≥2 questions with respect to the mean of the rest of the experts.

To validate both questionnaires, Aiken’s V coefficient values were calculated. For the questionnaires, we obtained a mean value of 0.89 (0.77–1.00) for nurses and 0.84 (0.77–0.95) for physicians. Results for all items of each questionnaire are shown in Table 6. The highest (≥0.9) Aiken’s V coefficient values were obtained for the items, number 1, 5, 8, 13, 19 in both questionnaires (i.e., Appendix A).

## 4. Discussion

The both choice-modeling Google-form questionnaires about self-perception of PAP barriers and facilitators by nurses and physicians were validated after a previous design following a rigorous method based on a content-analysis processing of two verbatim transcribed focus group sessions [12].

Item 1, focused on the preventive health benefits of PA and exercise, could discriminate if HC professionals do or do not promote PA, because of the lack of knowledge about the benefits or for other reasons still not resolved. However, the scientific literature provides clear evidence about the benefits of PA on human health [2,3,4]. PAP interventions in PCH settings have still a reduced evidence and need to be enhanced [41]. Previously, Desveaux et al. proposed a questionnaire to find facilitators and barriers of patients and HC providers in regard to a community-based exercise program [29], in a similar way than Myles et al., in Canada [36], Bock et al., in Germany [30], Freene et al., in Austalia [35], and Oyeyemi et al., in Nigeria [37]. Trying to filling in the gap, we propose a new validated questionnaire for analyzing feasibility of PAP in HC settings considering all dimensions and their respective integrated factors, according to the WHO 5 dimensions adherence model [16,42].

Item 5 was designed to assess the stage of change on PAP behavior for PHC professionals, according to the stage of changes of the transtheoretical model [43]. This question was introduced because many studies have evaluated the lifestyle behavior changes in patients or people [44,45], although currently to produce changes towards positive healthy lifestyles for a long-term time is not recognized as an efficient strategy in HC settings [45]. The participants of the previous focus group interview sessions [12] showed a lack of awareness as the majority of their professional colleagues. Because of this, the item tries to assess PAP behavior, based on the 5 stages of change behavior (pre-contemplation, contemplation, preparation, action, maintenance) in PHC providers. WHO proposed already in 2003 that health professionals need to be trained in non-pharmacological adherence treatments [16]. In order to improve adherence levels to PAP as a non-pharmacological treatment, to know what happens with patients is as important as to know about PHC staff. The panel of experts who validated the questionnaires seems to agree with this. Item 8 of the questionnaire was designed to assess the awareness on PAP by health professionals regarding the potential influence with their patients [24,25]. The experts agreed with the relevance of the question to be considered as an indicator in the self-perception barrier of PHC professionals in PAP. Item 13 asked about how PHC professionals were willing to collaborate in PA promotion through a multidisciplinary PHC team. This was scored as important by the experts, in line with the Exercise is Medicine initiative, which encourages multidisciplinarity [8,9]. Finally, item 19, related to the interest on PAP training courses for PHC professionals, was highly scored by all experts, being an indicator to enhance the efficient PAP implementation in PHC settings in accordance with previous studies [23,46,47]. Additionally, other training courses were demanded by the professionals, related to motivational interviews for PAP and time management and identified as resources needed to establish PAP efficiently in PHC settings, in accordance with other authors [48,49,50]. Therefore, this item was also included in the questionnaire and scored high by the experts.

This study could offer the key point to establish a validated questionnaire to analyze the main barriers of nurses and physicians at PHC settings to use PAP in their consultations under their self-perception, as a key factor, according to the social-ecological model [27]. In addition, knowing the positive predisposition of all PHC respondents, especially physicians, to collaborate with other health promoters and local community resources, these questions could be used to find the facilitators to design an effective PAP strategy to implement PAP in PHC settings, according to the health necessities and such as is proposed by recent European and global initiatives [6,51,52]. Limitations for this study were the equivalence on contents offered in the questions for nurses and physicians, according to the similitude of results in the content analysis of the semi-structured focus group sessions developed with them. This study has several strengths, as the validation procedure is based on a rigorous qualitative research process and the experts are highly prestigious in their fields. In a further study, a similar questionnaire process could be developed with exercise professionals in order to establish the self-perceived barriers and facilitators for an interdisciplinary PAP implementation team at HC settings.

## 5. Conclusions

The two 30 choice-modeling questions of the Google-form questionnaires have been validated by a panel of 10 experts and both are ready to be used in PHC systems to assess the self-perception of PAP facilitators and barriers of nurses and general practitioners. This should lead to facilitate the implementation of an efficient cost-effective and useful public health strategy in the HC system according to the Exercise is Medicine strategy.

## Figures and Tables

**Table 1 ijerph-17-06627-t001:** Summarized and justified structure of the questionnaire related to each category obtained by focus groups sessions.

CONSTRUCTS OF THE QUESTIONNAIRE				
Section	Category	Subcategory	No. Question	Observations
1.	Brief explanation			Summarized questionnaire presentation
2.	Personal and professional date			Personal and professional information
3.	Knowledge about PA benefits		1, 4	Objective basic knowledge about PAP to be considered accurate their following answers.
4.	Stage of change on PAP		5	Stage change on PAP behavior in PHC by nurses and physicians
5.	Self-perception Physical Activity (PA) pattern of Primary Healthcare (PHC) staff	Self-perception PA behavior of PHC professionals	6	To check the possibly influence in PAP an active behavior
	Physical Activity on Prescription (PAP) background	PAP knowledge and use	7, 8, 9, 10	Knowledge and use about PAP by nurses and primary physicians
	PHC staff position to PAP	Leadership to PA promotion	11, 12	Leader position in a PA promotion networking
		Leadership to exercise prescription	14, 15	Leader position in exercise prescription networking
		Collaborative PA promotion	13, 17, 18	Collaborative PA promotion networking
		Collaborative exercise prescription	16, 17, 18	Collaborative exercise prescription networking
	PAP training courses	PAP training courses status and necessities	19, 20, 21, 22, 23, 24, 25, 26	PAP training courses status and resource necessities in nurses and physicians of primary healthcare
	PAP as preventive and rehabilitation resource	PAP awareness by PHC Staff	8, 29.2	Use of PAP in the prevention and treatment of chronic diseases in PHC settings
		PAP Awareness by patients	29.6	Mass media advices to the patient by healthcare system
	PAP barriers	PAP anamnesis vital sign tool	27, 28	PA as vital sign in the health tool
		Improving consultation time management	29.3	Time to PAP in PHC consultation
		External policies relationships in PAP	29.5	PAP action plan by the healthcare system and external institutional relationship
		Lack of space resources	29.1	Lack of space to measure fitness and PA/sedentary levels
		Lack of Material-economic resources	29.4	Lack of economic and material resources
	PAP solutions	Modify PAP vital sign tool	30.1	Add an improved PA vital sign tool
		New space resources	30.2	Add a space to measure fitness and PA and possible PA training programs
		To create PAP networking	30.3	Develop a PAP networking
		To offer PAP Training courses	30.4	PAP Training courses to healthcare professionals
		PA advisement policies	30.5	PA advisement strategies in mass media
		Progressive PAP implementation in PHC	30.6	Progressive implantation of PAP in primary healthcare system
		To enhance material and economic resources for PAP	30.7	Add economic resources
		PAP leadership units at PHC	30.8	PAP leader position and structure in the PHC system
		Use of first consultation (nursering)	30.9	Use of first nurse consultation to measure physical fitness, PA levels, or something related to PAP
		Use of external PHC resources	30.10	Use of space outside PHC settings. i.e., walking routes, sports centers, etc.
		To increase PAP consultation time	30.11	Increase PAP consultation time because now is insufficient

**Table 2 ijerph-17-06627-t002:** Descriptive data of sports medicine physicians’ experts in physical activity on prescription (PAP).

Expert	Range Age	Sex	PAP Experience	Academic Background	Last University Studies Finished (Year)	Career Experience (Years)
1	51–55	Female	Researcher, University professor	Ph.D.	1985	20
2	51–55	Female	University professor, Healthcare professional	Ph.D.	1988	25
11 *	56–60	Male	Researcher, University professor, Healthcare professional	Ph.D.	1983	36
4	56–60	Male	Healthcare professional	Ph.D.	1982	31
5	>60	Female	Healthcare professional, Public Health promoter	Ph.D.	1977	36
6	56–60	Male	Researcher, University professor, Healthcare professional	Ph.D.	1986	30
7	> 60	Male	University professor, Healthcare professional	Master’s degree & Bachelor’s degree	1984	30
8	>60	Female	Healthcare professional	Ph.D.	1981	25
9	>60	Male	Researcher, University professor	Ph.D.	1981	25
10	51–55	Female	Researcher, University professor	Ph.D.	1989	26
Total		50% Male				28.4 ± 5.1

* Expert exchanged.

**Table 3 ijerph-17-06627-t003:** Descriptive data of nurses’ experts in physical activity on prescription (PAP).

Expert	Range Age	Sex	PAP Experience	Academic Background	Last University Studies Finished (Year)	Career Experience (Years)
1	56–60	Female	Healthcare professional	Bachelor’s degree	1983	35
2	46–50	Male	Researcher, University professor, Healthcare professional	Bachelor’s degree	1982	28
3	41–45	Female	Researcher, University professor	Ph.D.	2005	19
4	36–40	Female	University professor	Ph.D.	2003	10
5	41–45	Male	Healthcare professional	Master’s degree & Bachelor’s degree	1995	20
6	41–45	Female	Healthcare professional	Bachelor’s degree	1995	3
11 *	36–40	Female	Healthcare professional	Master’s degree	2004	4
8	46–50	Female	Healthcare professional, Public Health promoter	Master’s degree	2010	9
9	36–40	Male	Healthcare professional	Bachelor’s degree	2002	5
10	51–55	Female	Healthcare professional	Master’s degree	1982	30
Total		30% Male				16.3 ± 11.8

* Expert exchanged.

**Table 4 ijerph-17-06627-t004:** Values offered by the sports medicine physicians’ experts in all questionnaire items for the validation.

	Sports Medicine Physicians Questionnaire Validation										
	Expert 1	Expert 2	Expert 11 *	Expert 4	Expert 5	Expert 6	Expert 7	Expert 8	Expert 9	Expert 10	Expert 3 *
Item 1	5	5	5	4	4	5	5	5	5	5	5
Item 2	5	5	4	2	4	5	5	5	5	4	5
Item 3	5	4	3	4	3	5	5	5	5	3	3
Item 4	5	5	4	5	3	4	5	5	5	5	4
Item 5	4	5	5	5	3	5	5	5	5	5	5
Item 6	5	5	3	4	4	5	5	4	5	5	4
Item 7	4	5	5	4	4	3	5	5	5	5	5
Item 8	4	5	5	4	4	5	5	5	5	5	3
Item 9	3	4	3	3	4	5	5	4	5	5	4
Item 10	3	4	3	3	4	5	5	5	5	5	4
Item 11	4	3	3	4	4	4	5	4	5	5	4
Item 12	4	3	4	3	4	5	5	5	5	4	1 **
Item 13	3	4	5	5	4	5	5	5	5	5	3
Item 14	4	3	5	5	4	5	5	4	5	4	4
Item 15	3	3	5	5	4	5	5	5	5	4	1 **
Item 16	3	3	5	5	4	5	5	5	5	5	3
Item 17	3	4	4	5	4	5	5	5	5	5	4
Item 18	3	4	3	5	4	5	5	4	5	5	3
Item 19	3	4	5	5	4	5	5	5	5	5	5
Item 20	3	4	5	4	4	5	5	5	5	5	3
Item 21	4	3	3	4	4	5	5	5	5	4	4
Item 22	3	3	4	4	4	5	5	5	5	4	3
Item 23	3	4	3	4	4	5	5	5	5	4	3
Item 24	4	4	3	4	3	5	5	5	5	5	2
Item 25	4	4	3	4	3	5	5	5	5	5	3
Item 26	3	4	3	4	3	5	5	5	5	5	3
Item 27	4	4	5	5	4	5	5	3	5	5	1 **
Item 28	3	4	4	4	4	5	5	4	5	5	1 **
Item 29	3	5	4	4	4	5	5	5	3	4	2
Item 30	4	4	4	5	4	5	5	5	5	4	4

* Expert exchanged; ** values ≥ 3 ± SD of the mean of the rest of the 9 experts.

**Table 5 ijerph-17-06627-t005:** Values offered by the nurses’ experts in all questionnaire items for the validation.

Nurses Questionnaire Validation											
	Expert 1	Expert 2	Expert 3	Expert 4	Expert 5	Expert 6	Expert 11 *	Expert 8	Expert 9	Expert 10	Expert 7 *
Item 1	5	4	5	4	4	5	4	5	5	5	2
Item 2	5	4	3	3	4	5	4	4	4	5	1 **
Item 3	5	5	3	5	4	5	4	5	3	5	5
Item 4	4	5	4	4	4	5	4	5	5	5	4
Item 5	5	4	5	5	5	5	4	4	4	5	1 **
Item 6	5	4	3	4	4	5	3	5	5	5	4
Item 7	5	4	4	5	4	5	4	3	5	5	4
Item 8	5	4	5	5	5	5	3	4	5	5	3
Item 9	5	4	4	5	5	5	3	5	5	5	5
Item 10	5	4	4	5	4	5	4	4	3	5	5
Item 11	5	4	5	4	5	5	4	5	5	5	4
Item 12	4	4	4	5	5	5	3	5	3	4	2
Item 13	4	5	5	5	5	5	3	5	5	5	4
Item 14	5	5	4	5	5	5	4	5	5	5	2
Item 15	5	5	4	5	5	5	3	5	4	4	2
Item 16	5	5	5	5	5	5	3	5	4	5	4
Item 17	5	5	5	4	5	5	4	5	5	5	5
Item 18	5	5	5	5	5	5	4	5	5	5	5
Item 19	2	5	5	5	5	5	4	5	5	5	5
Item 20	5	5	5	5	5	5	4	5	5	5	5
Item 21	4	5	4	5	5	5	4	5	5	5	5
Item 22	5	5	5	5	5	5	5	5	5	5	5
Item 23	5	5	5	5	5	5	4	5	5	5	5
Item 24	2	4	4	4	5	5	3	5	4	5	5
Item 25	5	5	4	5	5	5	4	5	5	5	5
Item 26	2	5	2	4	5	5	3	5	4	5	5
Item 27	3	5	2	4	5	5	4	1	5	5	3
Item 28	5	5	3	4	5	5	4	2	5	5	4
Item 29	4	4	5	5	5	5	4	5	5	5	5
Item 30	5	5	5	5	5	5	4	5	5	5	5

* Expert exchanged; ** values ≥3 ± SD of the mean of rest 9 experts.

**Table 6 ijerph-17-06627-t006:** The mean and Aiken’s V coefficient score for all the thirty items offered by the ten experts.

	GPs’ Questionnaire							Nurses’ Questionnaire						
Item	1	2	3	4	5	Mean	Aiken’s V (95% CI *) Value (Range)	1	2	3	4	5	Mean	Aiken’s V (95% CI *) Value (Range)
1	0	0	0	2	8	4.80	0.95 (0.83–0.98)	0	0	0	4	6	4.60	0.90 (0.76–0.96)
2	0	1	0	3	6	4.40	0.85 (0.70–0.92)	0	0	2	5	3	4.10	0.77 (0.62–0.87)
3	0	0	3	2	5	4.20	0.80 (0.65–0.89)	0	0	2	2	6	4.40	0.85 (0.70–0.92)
4	0	0	1	2	7	4.60	0.90 (0.76–0.96)	0	0	0	5	5	4.50	0.87 (0.73–0.94)
5	0	0	1	1	8	4.70	0.92 (0.80–0.97)	0	0	0	4	6	4.60	0.90 (0.76–0.96)
6	0	0	1	3	6	4.50	0.87 (0.73–0.94)	0	0	2	3	5	4.30	0.82 (0.68–0.91)
7	0	0	1	3	6	4.50	0.87 (0.73–0.94)	0	0	1	4	5	4.40	0.85 (0.70–0.92)
8	0	0	0	3	7	4.70	0.92 (0.80–0.97)	0	0	1	2	7	4.60	0.90 (0.76–0.96)
9	0	0	3	3	4	4.10	0.77 (0.62–0.87)	0	0	1	2	7	4.60	0.90 (0.76–0.96)
10	0	0	3	2	5	4.20	0.80 (0.65–0.89)	0	0	1	5	4	4.30	0.82 (0.68–0.91)
11	0	0	2	5	3	4.10	0.77 (0.62–0.87)	0	0	0	3	7	4.70	0.92 (0.80–0.97)
12	0	0	2	4	4	4.20	0.80 (0.65–0.89)	0	0	2	4	4	4.20	0.80 (0.65–0.89)
13	0	0	1	2	7	4.60	0.90 (0.76–0.96)	0	0	1	1	8	4.70	0.92 (0.80–0.97)
14	0	0	1	4	5	4.40	0.85 (0.70–0.92)	0	0	0	2	8	4.80	0.95 (0.83–0.98)
15	0	0	2	2	6	4.40	0.85 (0.70–0.92)	0	0	0	3	6	4.50	0.87 (0.73–0.94)
16	0	0	2	1	7	4.50	0.87 (0.73–0.94)	0	0	1	1	8	4.70	0.92 (0.80–0.97)
17	0	0	1	3	6	4.50	0.87 (0.73–0.94)	0	0	0	2	8	4.80	0.95 (0.83–0.98)
18	0	0	2	3	5	4.30	0.82 (0.68–0.91)	0	0	0	1	9	4.90	0.97 (0.87–0.99)
19	0	0	1	2	7	4.60	0.90 (0.76–0.96)	0	1	0	1	8	4.60	0.90 (0.76–0.96)
20	0	0	1	3	6	4.5	0.87 (0.73–0.94)	0	0	0	1	9	4.90	0.97 (0.87–0.99)
21	0	0	2	4	4	4.20	0.80 (0.65–0.89)	0	0	0	3	7	4.70	0.92 (0.80–0.97)
22	0	0	2	4	4	4.20	0.80 (0.65–0.89)	0	0	0	0	10	5.00	1.00 (0.91–1.00)
23	0	0	2	4	4	4.20	0.80 (0.65–0.89)	0	0	0	1	9	4.90	0.97 (0.87–0.99)
24	0	0	2	3	5	4.30	0.82 (0.68–0.91)	0	1	1	4	4	4.10	0.77 (0.62–0.87)
25	0	0	2	3	5	4.30	0.82 (0.68–0.91)	0	0	0	2	8	4.80	0.95 (0.83–0.98)
26	0	0	2	4	4	4.20	0.80 (0.65–0.89)	0	1	2	2	5	4.10	0.77 (0.62–0.87)
27	0	0	1	3	6	4.50	0.87 (0.73–0.94)	0	1	2	2	5	4.10	0.77 (0.62–0.87)
28	0	0	1	5	4	4.30	0.82 (0.68–0.91)	0	1	1	2	6	4.30	0.82 (0.68–0.91)
29	0	0	2	4	4	4.20	0.80 (0.65–0.89)	0	0	0	3	7	4.70	0.92 (0.80–0.97)
30	0	0	0	5	5	4.50	0.87 (0.73–0.94)	0	0	0	0	1	4.90	0.97 (0.87–0.99)
Total							0.84 (SD ± 0.04)							0.89 (SD ± 0.06)

The Likert scale varied from 1 to 5, where the minimum (1) value is according to a very poor relevance and the maximum (5) to the highest degree of relevance. CI *: confidence intervals. GPs: general practitioners.

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
