# Peer review of "Design and Validity of a Choice-Modeling Questionnaire to Analyze the Feasibility of Implementing Physical Activity on Prescription at Primary Health-Care Settings"

_ijerph, 2020, doi:10.3390/ijerph17186627_

Round 1
Reviewer 1 Report
Congratulate them on a job well done. However, I would like to make a few comments on the document:
Correct quote tab number 6 and 44
Explain quote 12 why it appears at the end again
Table 6: Putting the word "mean" in the same line
Methodology section, show the type of study that has been carried out
Line 125: "The highest values (≥0.9) of Aiken's V coefficient were obtained for items, number 1, 5, 8, 13, 19".
Item 9,16,17,18,20,21 and others are missing
It would have been interesting to have the opinion as experts to professionals of the exercise, perhaps in limitations or future lines of research
Author Response
Dear Reviewer 1
First of all, we want to thank the reviewer 1 for these constructive comments. We have modified the article according to the suggestions and think that it has improved considerably.
According to the comment: Correct quote tab number 6 and 44
- Thank you very much for your suggestion. Now, we have corrected quote tab format of both references as you can see from line 229 to 231 and from line 338 to 340
According to the comment: Explain quote 12 why it appears at the end again
- Thanks. Following your comment, we would like to explain you that we have chosen quote 12 at the end of the references, because we understood in the rules of the IJERPH, that thesis references have to be published at the end in an independent thesis section. But now, we understand that it is not necessary, so we delete it. Please, let us know if it is published in a correct way.
According to the comment: Table 6: Putting the word "mean" in the same line
- Thank you for your comment. The word "mean" has been put in the same line of table 6
According to the comment: Methodology section, show the type of study that has been carried out
- Thanks for your comment of methodology section of our study. We have included the type of study that has been carried out.
According to the comment: Line 125: "The highest values (≥0.9) of Aiken's V coefficient were obtained for items, number 1, 5, 8, 13, 19". Item 9,16,17,18,20,21 and others are missing
- Thanks for your comment. In this statement we said "The highest (≥0.9) Aiken’s V coefficient values were obtained for the items, number 1, 5, 8, 13, 19 in both questionnaires" The highest values of items 9,16,17,18,20,21 are only for nurse´s questionnaire, not for both. For that reason, if the reviewer agrees, we would like to maintain the sentence in the same sense (from line 135 to 137).
According to the comment: It would have been interesting to have the opinion as experts to professionals of the exercise, perhaps in limitations or future lines of research
- Thank you for your suggestion. Definitively, we have introduced this statement "Limitations for this study were the equivalence on contents offered in the questions for nurses and physicians, according to the similitude of results in the content analysis of the semi-structured focus group sessions developed with them. This study has several strengths, as the validation procedure is based on a rigorous qualitative research process and the experts are highly prestigious in their fields. In a further study, a similar questionnaire process could be developed with exercise professionals in order to establish the self-perceived barriers and facilitators for an interdisciplinary PAP implementation team at HC settings" (From line 190 to 196).

Reviewer 2 Report
The introduction fails to explain 1) the significance of the study; 2) what has been done in the past; 3) what is the outstanding research gaps in terms of theory, practice and method. There is no explanation for how the questionnaire was developed, and how the validation study is going to proceed with the target standard of validity. The results look like raw data, not the results of analysis. Discussion does not make much sense because the study is not described properly. The entire manuscript needs to be rewritten by following a standard procedure of a quality study of this kind.
Author Response
Dear Reviewer 2
First of all, we want to thank the reviewer 2 for these constructive comments. According to your comments, we have thoroughly rewritten the manuscript. Furthermore, the English language and style of the article has been reviewed by a native English speaker. We think that it has improved considerably.
According to the comment: The introduction fails to explain 1) the significance of the study; 2) what has been done in the past; 3) what is the outstanding research gaps in terms of theory, practice and method…
- Regarding the introduction, in order to clarify the significance of the study, we have decided to delete the word "design" (line 66). Besides of clarifying what has been done in the past, we have introduced this new sentence " Some attempts have been done in the last years. Surveys have been designed and carried out in physicians from Canada [7, 29], Germany [30, 31] and USA [32, 33], in health promoters from Australia [34, 35] and physical therapists from Canada [36] and Nigeria [37] (from line 61 to 63). Furthermore, in order to clarify what are the outstanding research gaps in terms of theory, practice and method, we have written this new paragraph "… However, there is not a consensus or a validated questionnaire to measure the self-perceived PAP barriers and facilitators in health promoters. Therefore, as part of a broader study, the aim of this study was " (from line 64 to 66).
According to the comment: There is no explanation for how the questionnaire was developed, and how the validation study is going to proceed with the target standard of validity.
In the material and methods section, These new paragraphs have been introduced "Two questionnaires were validated according to the Guidelines published by Tsang et al., in 2017 [38]. Following these guidelines, domains of interest and constructs were previously identified to assess what the questionnaire should measure and a checking that no validated questionnaire is available in the scientific literature was performed. Baseline for the questionnaire design were focus groups discussions with 5 GPs and 5 nurses, which took place as part of a PhD Thesis [12]. Briefly, GPs and nurses, separately, guided by an expert psychologist, commented during four hours about their opinions, barriers, and facilitators to implement PAP in PHC. Their answers were categorized and organized in a structured 30-question questionnaire, according to the five sections and category and subcategory constructs shown in Table 1." (from line 71 to 79) and " Two choice-modeling, Google-form questionnaires were developed by our group [12] as an on-line self-administered format, because we thought that health professionals are able to complete the questionnaire on their own. A close-ended item format by multiple-choice, Likert-type scales and true/false answers were established, with a simple, short, and familiar language item design for the target respondents. As there is no rule for the length of questionnaires, it was designed to measure the full construct of interest, trying not be so long that respondents experienced fatigue or loss of motivation in completing it. A preliminary review and revision pool of items and pilot testing was conducted by all members of the imFINE research group.
Finally, a content validity of the questionnaires was done by a panel of experts as previously had been done by other authors in other fields [39, 40]." (from line 83 to 92). Besides, full details can be found in the reference number 12, cited in text and previously published by us.
According to the comment: The results look like raw data, not the results of analysis.
In order to enhance the results section of our manuscript, we have introduced this statement: " The 30items of each questionnaire were agreed by each expert from 1 to 5 points, being 5 the maximum agreed value. The sports medicine physician expert 3 (table 4) and the expert 7 of the nurses questionnaire (table 5) were not considered for offering 3±SD answers in ≥2 questions respect the mean of the rest of experts"; (from line 122 to 125). Besides of adding this improved statement "To validate both questionnaires, Aiken’s V coefficient values were calculated. For the questionnaires we obtained a mean value of 0.89 (0.77- 1.00) for nurses and 0.84 (0.77-0.95) for physicians. Results for all items of each questionnaire are shown in table 6. The highest (≥0.9) Aiken’s V coefficient values were obtained for the items, number 1, 5, 8, 13, 19 in both questionnaires (supplementary material)"; (from line 133 to 137).
Regarding the discussion, we have introduced the following statement " Previously, Desveaux et al., proposed a questionnaire to find facilitators and barriers of patients and HC providers in regard to a community-based exercise program [29], in a similar way than Myles et al, in Canada [36], Bock et al, 31 in Germany [30], Freene et al, in Austalia [35] and Oyeyemi et al, in Nigeria [37]. Trying to filling in the gap, ... " (from line 152 to 155) related to the new paragraph added in the introduction section in order to improve the relationship and sequence structure of the manuscript. In a similar way, we have added a new sentence "According to the item design related to the construct of interest, item 5 was designed…" from line 159, to restructure the new discussion section and improving the relationship with the new item designed explanation of materials and methods section.
Finally, these paragraphs have been added " In addition, knowing the positive predisposition of all PHC respondents, specially physicians, to collaborate with other health promoters and local community resources, these questions could be used to find the facilitators to design an effective PAP strategy to implement PAP in PHC settings, according to the health necessities and such as is proposed by recent European and global initiatives [51-53]. Limitations for this study were the equivalence on contents offered in the questions for nurses and physicians, according to the similitude of results in the content analysis of the semi-structured focus group sessions developed with them. This study has several strengths, as the validation procedure is based on a rigorous qualitative research process and the experts are highly prestigious in their fields. In a further study, a similar questionnaire process could be developed with exercise professionals in order to establish the self-perceived barriers and facilitators for an interdisciplinary PAP implementation team at HC settings." (from line 185 to 196). In order to improve the sense relationship with the aim and new added comments of our manuscript.

Reviewer 3 Report
The topic of your paper is interesting, and you provide some great background in the introduction to orient readers to the issues you explore and why those issues are important. I did encounter a few challenges with the article, as written, that I detail below:
1) The results section, as presented, was overwhelming. There is a lot of data in tables, but there is little offered in terms of context for readers to understand what the barrage of numbers actually means. It could be helpful to include some narrative descriptions to help readers make sense of the results to be able to better engage with the study and its meaning.
2) What was learned regarding the self-perception of PAP facilitators? While it seems the focus of the study was on validating the tools, the information gathered should tell some type of story that can be presented as a part of the manuscript.
Author Response
Dear Reviewer 3
First of all, we want to thank the reviewer 3 for these constructive comments. Furthermore, the English language of this article has been reviewed by a native English speaker. We think that it has been improved.
According to the comment: The topic of your paper is interesting, and you provide some great background in the introduction to orient readers to the issues you explore and why those issues are important.
- Thank you for your comment.
According to the comment: The results section, as presented, was overwhelming. There is a lot of data in tables, but there is little offered in terms of context for readers to understand what the barrage of numbers actually means. It could be helpful to include some narrative descriptions to help readers make sense of the results to be able to better engage with the study and its meaning.
- Thanks for your comments and suggestion to enhance the result section of our manuscript. In order to clarify it for readers, we have decided to introduce this statement: " The 30 items of each questionnaire were agreed by each expert from 1 to 5 points, being 5 the maximum agreed value. The sports medicine physician expert 3 (table 4) and the expert 7 of the nurses questionnaire (table 5) were not considered for offering 3±SD answers in ≥2 questions respect the mean of the rest of experts"; (from line 122 to 124). Besides, we have added this improved statement "To validate both questionnaires, Aiken’s V coefficient values were calculated. For the questionnaires, we obtained a mean value of 0.89 (0.77- 1.00) for nurses and 0.84 (0.77-0.95) for physicians . Results for all items of each questionnaire are shown in table 6. The highest (≥0.9) Aiken’s V coefficient values were obtained for the items, number 1, 5, 8, 13, 19 in both questionnaires (supplementary material)"; (from line 133 to 137).
According to the comment: What was learned regarding the self-perception of PAP facilitators? While it seems the focus of the study was on validating the tools, the information gathered should tell some type of story that can be presented as a part of the manuscript.
- Thank you for your question and your suggestion to improve our manuscript. As you commented, our main aim was to validate both questionnaires. Definitively, we have decided to introduce this paragraph: "In addition, knowing the positive predisposition of all PHC respondents, specially physicians, to collaborate with other health promoters and local community resources, these questions could be used to find the facilitators to design an effective PAP strategy to implement PAP in PHC settings…. " (from line 185 to 188), in the discussion section.

Round 2
Reviewer 2 Report
The significance of the study is not still justified by a sufficient logical argument. Just by reading the abstract, the reader wonders what the proper methodology that this study used.
Reviewer 3 Report
Thank you for addressing the comments I previously provided. The clarity of the manuscript is much improved.